# Dietary Advanced Glycation Endproducts and the Gastrointestinal Tract

**DOI:** 10.3390/nu12092814

**Published:** 2020-09-14

**Authors:** Timme van der Lugt, Antoon Opperhuizen, Aalt Bast, Misha F. Vrolijk

**Affiliations:** 1Department of Pharmacology and Toxicology, Maastricht University, 6229 ER Maastricht, The Netherlands; antoon.opperhuizen@maastrichtuniversity.nl; 2Office for Risk Assessment and Research, Netherlands Food and Consumer Product Safety Authority (NVWA), 3540 AA Utrecht, The Netherlands; 3Campus Venlo, Maastricht University, 5911 BV Venlo, The Netherlands; a.bast@maastrichtuniversity.nl (A.B.); m.vrolijk@maastrichtuniversity.nl (M.F.V.)

**Keywords:** inflammatory bowel disease, dietary advanced glycation endproducts, gastrointestinal tract, inflammation, digestion

## Abstract

The prevalence of inflammatory bowel diseases (IBD) is increasing in the world. The introduction of the Western diet has been suggested as a potential explanation of increased prevalence. The Western diet includes highly processed food products, and often include thermal treatment. During thermal treatment, the Maillard reaction can occur, leading to the formation of dietary advanced glycation endproducts (dAGEs). In this review, different biological effects of dAGEs are discussed, including their digestion, absorption, formation, and degradation in the gastrointestinal tract, with an emphasis on their pro-inflammatory effects. In addition, potential mechanisms in the inflammatory effects of dAGEs are discussed. This review also specifically elaborates on the involvement of the effects of dAGEs in IBD and focuses on evidence regarding the involvement of dAGEs in the symptoms of IBD. Finally, knowledge gaps that still need to be filled are identified.

## 1. Introduction

Inflammatory bowel diseases (IBD) is the most commonly used term for a group of chronic inflammatory conditions of the gastrointestinal tract, of which ulcerative colitis (UC) and Crohn’s disease (CD) are the two main types. Over the past years, these inflammatory diseases have emerged as a public health challenge. The prevalence of IBD has been increasing substantially over the last decades. In 2017, approximately 6–8 million people suffered from IBD [1]. Although the aetiology of IBD has been extensively investigated, the exact pathogenesis is still not fully known. Multiple factors are involved in the pathogenesis of IBD, such as environmental factors, immunological factors, genetic susceptibility, and diet [2]. Not only in the Western world, but also in newly industrialised parts of the world such as Brazil and Taiwan, the incidence of IBD is increasing [3]. This suggests the important role of the environment in the development of the disease. More specifically, the introduction of the Western diet in those areas has been suggested as a potential explanation of increased IBD incidence [1,4]. The Western diet typically includes high levels of sugars and fats and is usually highly processed. During food processing, new compounds are introduced to food and process contaminants are being formed. Additionally, food products are often enriched with sugars and fats; proteins are often added to obtain foods that may help build muscle. All of these can potentially impact human health. The processes that food products undergo often include industrial heat treatments, such as sterilization, but also cooking and baking at home. During thermal treatment of foods which contain proteins and carbohydrates, the Maillard reaction (MR) can occur. This is a chemical reaction between reducing sugars and amino acids that is responsible for the browning of food. The MR is an intricate reaction with multiple parallel and sequential steps, as presented in Figure 1, during which many different compound groups are formed. These compounds are also known as MR products (MRPs), which include Amadori products, early glycation products, advanced glycation endproducts (AGEs), and melanoidins. Especially AGEs have gained a lot of attention in recent years due to their potential negative effect on human health. Existing literature on AGEs mostly focuses on endogenously formed AGEs, which form when endogenous sugars react with endogenous proteins at 37 °C. The health effects of endogenous AGEs, including chronic inflammation, tissue damage, stiffening of collagen, and enhancement of oxidative stress, have been studied thoroughly [5]. In addition to the endogenous formation, AGEs can also be formed in food during processing. These AGEs are then considered as dietary AGEs (dAGEs). Differences in the health effects between endogenous AGEs and dAGEs are expected due to the different manner of formation between both: Endogenous AGEs are formed at 37 °C with endogenous carbohydrates and endogenous proteins over a longer period of time, while dAGEs are formed with exogenous proteins and carbohydrates in a shorter period of time and under more extreme conditions, e.g., higher temperatures. Consequently, different subsets of molecules are expected to be formed. This was also shown in a recent review by Liang et al. (2019), who summarized the occurrence of different AGEs in food and in the body. In contrast to the endogenous AGEs, dAGEs have been less well studied and several questions about these dAGEs remain, including what their effects may be, if and how they are absorbed in the human intestinal tract. The latter is especially important since absorption of these dAGEs is necessary in order to have systemic effects. Some excellent and thorough reviews have already been published on the systemic effects of dAGEs [6,7,8,9]. In contrast, it is still disputed, to what extent and in what form, dAGEs are absorbed. Independent of absorption, dAGEs enter the human gastrointestinal (GI) tract after ingestion. Studies into the local effects of dAGEs in the GI tract are scarce and it is therefore not known what the local effects of dAGES in the GI tract are. Our research group recently found that dAGEs evoke an inflammatory response in vitro. Due to these potential pro-inflammatory effects of dAGEs, it can be questioned whether dAGEs could be involved in the inflammatory responses for instance in IBD patients. Therefore, the present review focusses on the effects of dAGEs in the GI tract and identifies knowledge gaps that still need to be filled.

## 2. The Maillard Reaction

The MR is an intricate reaction with multiple parallel and sequential steps, as presented in Figure 1. In the first initial stage, a carbonyl (reducing sugars, oxidized lipids, vitamin C or quinones) condensates with an amine moiety on amino acids and proteins, which leads to the formation of a Schiff base adduct. Following this stage, the Schiff base adduct is rearranged into an Amadori product in the intermediate stage. Amadori products are unstable and prone to polymerization due to being highly unsaturated. In the final stage, both sequential and parallel reactions lead to different molecules. These reactions include oxidation, dehydration, enolization, cyclization, and fragmentation [10]. During the MR, many different classes of compounds (MRPs) are formed. Especially the subgroup AGEs, which are formed during one of the last stages of the MR have received attention lately. The term AGEs implies a whole group of numerous different molecules. Some well-known AGEs are: N^ε^-carboxymethyl-lysine (CML), N^ε^-carboxyethyl-lysine (CEL), methylglyoxal-derived hydroimidazolone-1 (MG-H1), pentosidine, and pyrraline. Within this large group of different AGEs, it is possible to make a distinction between low molecular weight (LMW) and high molecular weight (HMW) AGEs. LMW AGEs consist of up to four linked rings of up to 1 kDa [11]. HMW AGEs are assumed to be water-soluble and coloured, weighing up to 150 kDa [11]. Additionally, AGEs exist in both a protein-bound form and a free form and their visible colour is brown. The formation of AGEs has been nicely explained in the review by Poulsen et al. (2013) [10]. AGEs do not only form during the MR, since also other reactions may lead to the formation of AGEs. Intermediate compounds that can form AGEs are 1,2-dicarbonyl derivatives such as glyoxal (GO), methylglyoxal (MG), and 3-deoxyglucosone (3-DG), which can interact with monoacids to form AGEs. MG and GO can be formed during caramelization, while through the radical Namiki pathway (lipid oxidation) 1,2-dicarbonyls can also be formed [5]. Fructose metabolites such as fructose-3-phosphate are formed in the polyol pathway, which are then converted into α-oxaldehydes. α-Oxaldehydes can form AGEs by interacting with monoacids [7,12]. The formation of MRPs can be monitored by the formation of brown colours, as Hofmann et al. (1998) stated that the browning intensity seemed to be parallel with cross-linking of casein [13].

## 3. Terminology

As mentioned earlier, Amadori products, early glycation products, AGEs, and melanoidins all fall under the umbrella term ‘MRPs’. Hence, all these compounds are being formed during thermal processing of food rich in carbohydrates and proteins. Therefore, a problem that arises with studies into the effect of dAGEs when using food matrices, is the simultaneous formation of this broad and varying subset of molecules and the unclear border between these different subsets. dAGEs have defined molecular structures such as CML, CEL, MG-H1, pentosidine, and pyrraline. A second subset of molecules formed in the end-stage of the MR are melanoidins, which have a higher molecular weight than dAGEs. The dAGEs are considered to be the reactive intermediates, whose condensation forms melanoidins [14,15]. The use of the terms ‘AGEs’ and ‘melanoidins’ in existing literature is not always fully clear. dAGEs contribute to the brown colour of thermally treated food and some of them have fluorescent properties. Remarkably, melanoidins are also defined as brown, high molecular weight heterogeneous polymers, and are usually analysed in food products by measuring the intensity of the browning and fluorescence [14]. Therefore, it is almost impossible to out rule the presence of dAGEs in those studies measuring melanoidin levels only.

The presence and ratio of the different subsets in food products is dependent on the food matrix and the above discussed processing conditions. The difficult distinction within the end-stage of the MR makes it difficult to attribute the effects of heated foods to specific compounds. In this review, we will try to distinguish between the different subsets and focus on the effects of dAGEs as a subset but will include the effects of MRPs in general since all of these compounds are present in food, but attempt to make a distinction in the terminology where possible. When discussing the literature, it will be specifically stated which subset of compounds were investigated.

## 4. Dietary AGEs in the Gastro-Intestinal Tract: Digestion, Absorption, Formation, and Degradation

In the determination of the local gastro-intestinal and systemic effects of dAGEs, it is important to know how digestion affects the availability of dAGES. After the ingestion of thermally processed food products, dAGEs enter the human GI tract. Since proteins are vulnerable to the acidity of the stomach, it is to be expected that at least the protein-dAGE-binding will be altered. The following sections will explore how dAGEs are digested and absorbed and how new AGEs are formed in the GI tract.

### 4.1. Digestion of AGEs

Studies into the effect of gastric digestion on dAGEs are scarce and most studies are conducted using static in vitro models. In these models, food products are most commonly exposed to synthetic gastric juice and pepsin set at a pH value of 2–2.5 for two hours. However, human gastric digestion is not a static environment; it is dynamic and includes peristaltic mixing and alternating pH levels [16]. The physiological pH levels in the stomach differ between 1–1.5 for basal levels and 5–7 immediately after ingestion. After approximately three hours of ingestion, the pH returns to basal levels [17]. Additionally, digestive fluids and meal components are removed from the GI tract by both passive and active absorption [16].

In order to determine the effect of gastric digestion on dAGEs, Hellwig et al. (2013) exposed casein modified with N^ε^-fructoselysine, N^ε^-carboxymethyllysine, and lysinoalanine to synthetic gastric juice and pepsin with a pH value of 2 at 37 °C of 2 h. In this study, modifying casein led to a reduction of digestibility of casein in the intestines [18]. Although most of the physical parameters were taken into account, all dAGEs were exposed to the gastric juices for the complete two hours. This is not the case in the human body where constantly a small portion of the meal is transported to the duodenum and gastric emptying time is depending on the type of food [19,20]. In addition, several other studies have been published on the digestibility of dAGEs in the small and large intestine. Joubran et al. (2015) assessed the digestive proteolysis of glycated α-lactalbumin in adults and infant gastro-duodenal conditions [21]. The type of carbohydrate used in the experiments determined the extent to which proteolysis could occur. Digestion of glycated α-lactalbumin led to higher anti-oxidative properties, while other oligosaccharides protected against proteolysis. However, the details on what gastric conditions were used, were not disclosed in the paper and their method of thermal processing corresponded to neither endogenous nor exogenous conditions. Underlining these results, Zhao et al. (2017) saw a decrease in digestibility of β-casein and β-lactoglobulin after glycation with glyoxal [22]. In a different study, Pinto et al. (2013) subjected β-lactoglobulin and β-casein to a heating regime of 90 °C for 24 h and found larger aggregates of casein combined with glucose than native casein, while glycation retarded the aggregation of β-lactoglobulin. Moreover, when heated both glucose-protein combinations were less sensitive to enzymatic digestion than their heated native counterparts [23]. Moscovici et al. (2014) found a decreased sensitivity of bovine lactoferrin glycated with fructo-oligosaccharides to proteolysis after heating at 60 °C in a water-restricted environment for 12 h, and an enhanced proteolysis after 24 h of heating [24]. Both studies indicate the importance of the starting product and thermal treatment on the effect on tertiary structure of the protein and thereby its digestibility.

Unfortunately, the static in vitro models as used by the above discussed studies do not take such physiological parameters into account. Since these studies all used static in vitro models, we conducted a recent study using a sophisticated in vitro GI model to assess the digestion of dAGEs and their protein binding [25]. The TNO gastroIntestinal Model-1 (TIM-1) is a dynamic in vitro model which mimics the upper human GI tract. This model takes multiple parameters into account, such as: Dynamic pH curves, peristaltic mixing, addition of bile and pancreatic digestive enzymes, and passive absorption of fluids and food compounds. We analysed both our food-based matrix (GC) and two actual food products: Ginger biscuits and apple juice. Our results showed that the protein binding of dAGEs is able to survive gastric and small intestinal digestion and consequently stays intact throughout the whole GI tract. We also found a hampered digestion of GC compared to UC.

In addition to the above mentioned experimental studies, an excellent systematic review by van Lieshout et al. (2019) analysing the effects of glycation on protein digestibility concluded that glycation decreases protein digestibility [26]. One explanation for the reduction of protein digestibility by the MR lies in the nature of the reaction itself. The major reactive site for the MR is the ε-amino group of lysine, and to a lesser extent the guanido group of arginine [27]. These same residues are also tryptic binding sites [28]. Consequently, this leads to a direct hindrance of the proteolytic cleavage sites [28,29]. Additionally, indirect steric hindrance of the proteolytic cleavage sites can occur through binding to residues close to proteolytic cleavage sites, or it can cause cross-linking that inhibits access for proteases [26]. Alternatively, glycation may also promote proteolysis by opening up the tertiary structure of proteins (conjugation may interfere with refolding the protein after heating [29]), which was observed in the study by Moscovici [24]. A recent human trial administering milk protein to healthy young men, which was glycated to different extents, found a reduced lysine availability that was dose-dependent on the level of glycation. This reduced availability was only seen for lysine, not for the other amino acids [30]. Additionally, Joubran et al. (2017) found that the MR changes the functionality of the whey protein α-lactalbumin. They also found that the specific carbohydrate involved in the reaction determined the change of the protein [29].

Based on these studies, it can be concluded that dAGEs themselves and their protein binding are not broken down in the GI tract. The extent of digestion is furthermore depending on the raw materials and heat treatment of dAGE-containing foods. Additionally, the binding of dAGEs to proteins leads to a decreased digestibility and bioavailability of amino acids, with lysine in particular.

### 4.2. Absorption

In order to be able to assess the systemic and local effects of dAGEs, it is important to know whether dAGEs are absorbed. When dAGEs are not absorbed, they will remain in the GI tract, travel further, and potentially affect the colon. A few reviews have already included an assessment of the absorption of specific dAGEs: Delgado et al. (2015) concluded that CML is taken up to some extent. More specifically, free CML was most likely absorbed by simple diffusion and CML bound to dipeptides by peptide transporters (PEPT1) [31]. Faist et al. (2001) and Somoza et al. (2005) both concluded that MRPs are most likely taken up, but that more research on the uptake and mechanisms is needed to confirm this. In general, HMW dAGEs may be absorbed to a much lesser extent than LMW dAGEs [32,33]. In addition to these reviews, several in vitro studies on the absorption mechanism of specific AGEs have been published and their results are summarized in Table 1. Most free dAGEs are capable of entering the GI cells by diffusion, but seem to be retained inside the cell, whereas peptide-bound dAGEs are transported inside the cell by peptide transporters.

The absorption of dAGEs has also been investigated in several in vivo animal studies. It should be noted that the physiological relevance of such animal studies regarding the uptake in humans is debatable, due to large differences in GI physiology between the animal species and humans. Taking the available animal studies into account, differences between free compounds and protein bound compounds exist in their ability to be absorbed. When assessing the different animal studies, free compounds seem to be taken up more easily [10], although this seems not to be the case for free CML [38]. HMW dAGEs are less absorbed due to insufficient degradation by GI enzymes. Intragastric administration of ^18^F- CML in mice showed a 48% accumulation of CML in the intestines, 29% in the bladder, and 18% in the stomach 120 min after administration [39].

Several human studies saw correlations between dAGE intake and serum AGE levels [40,41,42]. An intervention study with a low AGE diet showed that changes in serum CML and the AGE precursor MG significantly correlated with changes in dAGE levels in the diet [43]. This is also seen in infants fed with infant formulas (IF), which contain more CML than human breast milk. Infants solely fed with IF had higher levels of CML in their blood and urine than breast-fed infants [44]. Additionally, urinary excretion of pyrraline, fructoselysine (both 90%), and pentosidine (40%) decreased while consuming a diet low in dAGEs [42]. A different study by Davis et al. (2015) exposed overweight and obese adults to a crossover exposure of one day to high-fat, high-AGE and another day to low-fat, high-AGE diet. These acute exposures showed no important influence of dietary CML on CML serum levels, but rather an effect of high-fat diet compared to low-fat diet on CML serum levels [45]. A recent study by Scheijen et al. (2018) assessed total (no distinction between free and protein bound) dietary CML, CEL, and MG-H1 intake of 450 people with an elevated risk for T2DM and cardiovascular disease. A correlation between the intake of all three dAGEs and their free form levels in plasma was found. However, this correlation was not found for the protein-bound plasma levels of the different dAGEs [46]. Based on this, it can be concluded that free dAGEs are indeed taken up in the systemic circulation. Overall, it must be noted that these studies gave no clear conclusion on the amounts or percentages of uptake of dAGEs. Hence, more research is needed to the extent and the exact mechanism of uptake. Since in the aforementioned study by Davis et al. (2015) dAGEs seem to be better absorbed during a high fat diet, we also propose micellar uptake as an interesting mechanism for future studies.

### 4.3. Formation of AGEs in the GI Tract

In addition to not being hydrolysed in the GI tract, evidence exists on the endogenous formation of AGEs within the GI tract [47]. After in vitro gastric and small intestinal digestion of different food products in our study using the TIM-1 model, we saw in increase of almost 400% total (free and protein bound) MG-H1, 300% G-H1, and more than 200% CML and CEL content in the small intestinal digests of ginger biscuits. Additionally, an increase of almost 200% free form MG-H1 in the digests of apple juice. This formation was both dependent on the type of food products and thermal treatment, as the unheated glucose-lactose-casein meal system did not lead to any intestinal AGE formation [25]. Other studies have also shown intestinal AGE formation: Simulated digestion of ovalbumine with fructose alone produced compounds with a fluorescence spectrum that is characteristic for AGEs [48]. The circumstances under which intestinal AGE formation takes place are not fully understood yet. DeChristopher suggested that the formation of fructose-derived AGEs in the small intestine can be explained by fructose malabsorption. This may consequently, lead to increased serum AGE levels [49,50]. Fructose malabsorption can occur when the ratio of fructose to glucose is higher than 1:1, or when an individual has a fructose transporter deficiency. Normally, fructose is passively transported in the intestinal cell by glucose transporter 5 (GLUT5) and transported into the bloodstream by glucose transporter 2 (GLUT2) [51]. However, evidence suggests that glucose enhances fructose absorption in humans possibly through co-absorption via GLUT2 [52,53]. Ratios higher than one can be found in, for instance, high fructose corn syrup and apple juice. As a consequence of fructose malabsorption, its levels will increase inside the intestines, which might eventually lead to the increased formation of AGEs due to the more reactive nature of fructose in the MR than glucose. Although fructose-specific AGEs are not well defined, fructose seems to be 7–10 times more sensitive to glycation than glucose [48,54]. As stated earlier, conflicting evidence exists on the correlation between dAGE intake and endogenous AGE levels.

Although many of these studies focused on fructose-based dAGEs. The effect of glucose on intestinal AGE formation was also shown by Martinez-Saez in an unheated meal resembling system [47]. Helou et al. (2017) proposed a different hypothesis for the formation of AGEs in the GI tract. In their study, they used the TIM-1 model to assess the digestion of fluorescent soluble melanoidins/AGEs from bread crust, and found an increase in fluorescent molecules in the digests [55]. The authors concluded that, since the TIM-1 model only has a temperature of 37 °C, the increase was related to the release of soluble melanoidins from larger insoluble melanoidin-skeletons. This would mean that proteins conceal glycated parts that are released during glycation. However, our study included a hydrolysation step in which the dAGEs were detached from their proteins, circumventing this problem. The data showed that new AGEs are being formed in the intestine and not necessarily released from their structures. In addition, AGEs were only formed endogenously after digestion of products that also contained AGEs, which might indicate an underlying need for the presence of dAGE precursors in food, such as glyoxal and methylglyoxal [25]. To our knowledge, no studies have focused on the role of dAGE precursors on AGE formation in the GI tract and this might be an interesting topic for future research. Additionally, it is unknown how long dAGEs remain in the GI tract; the retention time could possibly influence the extent to which dAGEs are formed. However, we do hypothesise that this will depend rather on the food matrix than the dAGEs themselves.

### 4.4. Degradation and Effect in the Colon

Since dAGEs are not fully digested and absorbed in the GI tract, they may be further transported into the colon where they come into contact with the human colonic microbiota. Based on the available literature, the human colonic microbiota seems to be capable of only fermenting LMW dAGEs and not the more complex HMW structures, including dAGEs and melanoidins. Xu et al. (2019) subjected glyoxal-glycated casein to static in vitro digestion and assessed the capability of human colonic microbiota to ferment these digested compounds [56]. Their results showed that human colonic microbiota was indeed able to ferment the LMW structures to a better extent than HMW structures. Although in a different study *Bifidobacteria* have been shown to be capable of using bread melanoidins as a carbon source, indicating the prebiotic potential of bread melanoidins. However, the melanoidins were not digested in a digestion model prior to microbial exposure [57]. For the LMW weight structures, additional evidence can be found on the utilization of Amadori products, or early glycation products, by colonic microbiota. Human intestinal microbiota contain deglycating enzyme systems for Amadori products. For instance, *E. coli* has genes encoding for fructosamine kinase, which catalyses the production of the stable intermediate fructosamine-6-phosphate, and fructosamine-6-phosphate deglycase that in turn cleaves fructosamine-6-phosphate to glucose-6-phosphate and the corresponding amines [58]. Additionally, *E. coli* has been shown to utilize fructoselysine, and a human gut commensal (*Intestinimonas* AF211) can produce butyrate from lysine and fructoselysine [59,60]. Amadori products can furthermore be degraded to 1,2-dicarbonyl compounds by colonic bacteria, which are highly reactive with lysine ε-amino groups and arginine guanidine groups to form AGEs within the colon [34].

Due to the decreased digestibility of glycated proteins, unabsorbed amino acids and proteins are also transported into the colon. Protein fermentation in the colon is often detrimental due to the formation of several products that negatively affect the colon such as ammonia, amines, phenols, and sulfides [61,62]. As a consequence of dAGE fermentation by the microbiota and the accompanying produced compounds, the composition of the microbiota and its production of short-chain fatty acids (SCFA), which are generally considered anti-inflammatory, can change [63,64]. A recent review by Snelson et al. (2019) assessed the effects of dAGEs on gut microbiota regarding composition changes and SCFA production [65]. Concerning the microbiota composition, conflicting results between the in vitro and animal in vivo studies were found. However, their conclusions were that in animals, high-dAGE diets were associated with a decrease in Bateroidetes, *Bifidobacteria, Lactobacilli*, and α diversity (the number of different species). It is generally considered that a decrease in all these three species leads to a less healthy GI tract. The effects of dAGEs on the microbiota seem to be dependent not only on the type of food or glycated material used, but also on biological differences between individuals [55]. In general, human in vivo studies assessing the effect of dAGEs on microbiota are scarce. Most studies on the effect of dAGEs on the microbiota used one type of food product or glycated material and one type of heating, which is not an optimal representation because the treatment and starting product largely influence the outcome. This was additionally shown in a recent in vitro study by Pérez-Burillo et al. (2018) who analysed the effect of different food products (chickpeas, bread, red pepper, banana, and chicken) on SCFA production [66]. All food products were prepared in different ways (i.e., roasted, fried, boiled, and grilled). The food products were digested with a static in vitro GI digestion model and consecutively fermented in vitro. After in vitro fermentation, SCFA production was analysed as well as the microbial community composition. Their main findings confirm that the effects on the microbiota both greatly depended on the type of food product and the cooking technique. It must be noted that the analyses of the food products focused on furosine (early glycation), and 5-(hydroxymethyl) furfural (HMF) (intermediate glycation), so the composition of specfic dAGEs in the food products was not measured.

Overall, dAGEs do seem to influence the composition of the intestinal microbiota and its functionality. However, the effects are largely dependent on food composition and thermal treatment, as was also seen for the formation of intestinal AGEs. Therefore, more research is needed to draw any firm conclusions.

## 5. Local Health Effect in the Intestinal Tract

Based on the aforementioned findings that dAGEs are absorbed to only a small extent and that dAGEs enter in both free and protein bound form the colon, we suggest that the potential local effect of dAGEs in the GI tract may be of larger importance than those of dAGEs that end up systemically.

### 5.1. Pro-Inflammatory Effect of dAGEs

Animal and human studies regarding the potential pro-inflammatory effect of dAGEs on general human health have been reviewed several times. Poulsen et al. (2013) concluded that in animals fed a high dAGE diet, expression of the specific receptor for advanced glycation endproducts (RAGE) increased in several tissues [10]. Van Puyvelde et al. (2014) found that the dietary intake of dAGEs is significantly correlated with inflammatory processes and oxidative stress. In humans, a relationship is seen between a low-dAGE diet and suppression of inflammatory parameters. However, no clear increase is seen when given a high-dAGE diet [67,68]. One possible confounder in high-dAGE versus low-dAGE diets, is that generally low-dAGE diets have less calories than high-dAGE diets. Therefore, beneficial effects of a low-AGE diet may be caused by calorie-restriction rather than dAGE-restriction. However, animal data also suggest that the positive outcome of calorie-restriction might be due to a decreased dAGE-content. Several studies show a detrimental effect of heated diets in rodents compared to the unheated equivalent in isocaloric diets [69,70]. However, as was pointed out by Buetler et al. (2009), many essential diet components (such as vitamins), are heat-liable, and heating the rodent diet possibly destroyed these compounds, also leading to a detrimental effect [71,72]. It must also be noted that experiments with heat-processed food in animals are not very representative for the effect on human health, due to differences in the GI tracts and that animals do not have access to heated food products in their natural environment. This indicates that properly designed clinical trials are needed to investigate the possible interplay between calorie-restriction and low dAGE-diets regarding the effect on human health.

Studies into the pro-inflammatory effect of dAGEs in the healthy intestinal tract are practically non-existent, and most in vitro studies are performed in macrophages. Studies exposing macrophages to dAGEs do show a pro-inflammatory outcome [73,74,75]. One aspect much debated is whether free, individual dAGEs cause a pro-inflammatory effect or only the protein-bound dAGEs. As discussed before, only protein- or peptide-bound dAGEs seem to bind to RAGE. Evidence regarding the pro-inflammatory effect of free dAGEs is scarce. Acrylamide is also formed via the MR from asparagine and is considered to be a carcinogenic compound [76]. This compound is very extensively studied, but studies on the pro-inflammatory potential are non-existent. The evidence for the pro-inflammatory effects of protein bound dAGEs is more abundant. We have showed in human macrophage-like cells, protein bound dAGEs induced TNF-α secretion whereas individual, free dAGEs did not [73]. In HUVECs and mouse peritoneal macrophages, dAGEs bound to BSA induced an inflammatory response [74]. The results indicated as well that AGE receptors do not seem to be the sole mediator of the inflammatory response. Additionally, in a second study they showed that the glycated whey protein concentrate induced phagocytic activity of RAW264.7 cells [75]. However, no endotoxin test was performed for the samples in the second study, making the results less reliable.

### 5.2. Other Local Effects

Over the last decades, the prevalence of immune-mediated diseases, such as allergies, has increased drastically [77,78]. A main cause for this increase has not been found yet. An increasing amount of evidence suggests a role of dAGEs, especially in food allergy [79]. An elegant review by Teodorowicz et al. (2017) listed the mechanisms by which the MR may influence the allergenicity of proteins. It was concluded that the MR could reduce allergenicity through blocking of epitopes in proteins, or can enhance allergenicity through the formation of new epitopes or the formation of agglomerates that increase IgE binding [80]. Again, the heterogeneity of dAGEs is most likely responsible for the different outcomes associated with allergenicity potential. Therefore, more research is required to elucidate the effects of dAGEs on the effects on food allergy.

In addition to these immunological effects, MRPs are proposed to have anti-oxidative properties based on a large body of in vitro scientific literature. A substantial amount of these studies is on the effect of melanoidins. As explained earlier, during the thermal treatment of food products, all different MR products are being formed and the presence of dAGEs cannot be excluded. Therefore, it would not be appropriate and correct to exclude the effects of these compounds from this review.

Yáñez et al. (2018) analysed the effect of the rate of MR on the formation of antioxidants during the reaction through the measurement of DPPH radical scavenging capacity of MRPs. Additionally, a faster MR rate (in the presence of glucose) led to a lower anti-oxidative capacity than when the MR progressed at a slow rate (in the presence of lactose). An explanation for the higher antioxidant formation can be the difference in chemical structure between lactose and glucose, leading to the formation of different compounds through the MR.

In a different study aimed to analyse the anti-oxidative capacity of MRPs, biscuits were heated for different amounts of time, after which they were digested using a static in vitro digestive model duration [81]. Subsequently, the digests were incubated with human faeces after which DPPH, FRAP, and IC_RED_ were measured. The results show increased anti-oxidative capacity after digestion and fermentation with faeces at higher preparation temperatures and longer durations [81]. The following in vivo study in which WT Wistar rats were given diets differing in the MRP content showed possible probiotic activity for the diets high in MRPs.

One of the major whey proteins in milk and infant formula is α-lactalbumin. This protein possesses several biological properties among which serving as an antioxidant. Joubran et al. (2017) assessed the effect of glycation of α-lactalbumin with several carbohydrates (glucose, galactose, and galacto-oligosaccharides) on its anti-oxidative properties and saw an increase in anti-oxidative capacity of glycated α-lactalbumin after in vitro digestion [29]. Chuyen et al. (1998) attempted to measure the anti-oxidative capacity of specifically Amadori products and melanoidins and observed anti-oxidative properties by both compound groups, with a larger capacity when the molecular structure of the melanoidins were larger [82]. We conclude that the compounds formed during the MR have diverse effects on human health, form pro-inflammatory to anti-oxidative. We propose that the ratio of different subsets formed might determine the eventual outcome of heated food products.

In conclusion, based on the available literature, dAGEs do seem to exert several effects on intestinal health: Activating macrophages and changing the protein structures leading to a possible different allergenicity potential. However, dAGEs also seem to have anti-oxidative effects. Therefore, more research is needed to elucidate these different effects and to determine the most important and predominant effects. Additionally, the heterogeneity of MRPs should be taken into account in future studies.

## 6. Mechanisms

Although human clinical evidence on inflammatory effects of dAGEs in the GI tract remains scarce, several reviews exist about the general risks of AGEs for human health. Therefore, we will only give a brief overview about the general molecular mechanisms of AGEs under (patho)physiological conditions, including the receptor mediated and non-receptor mediated effects.

### 6.1. Receptor Mediated Effects

AGEs can bind to different receptors, of which RAGE is the most well-known. Under normal physiological conditions, RAGE is expressed at low levels in all organs, except in lung tissue, which has a high baseline expression of RAGE [83]. Under pathophysiological conditions, such as diabetes, inflammatory diseases, and neurodegenerative disorders, RAGE is strongly upregulated in different tissues such as vasculature, central nervous system, and GI tract [83,84,85]. RAGE is a 35 kDa transmembrane receptor and member of the immunoglobulin superfamily, a large protein family that shares structural features with antibodies [86]. Different variants of RAGE exist, of which full length RAGE is the only variant that is both capable of ligand binding and signal transduction. Full length RAGE consists of the extracellular site of a V-type domain for ligand binding and two C-type domains of which their exact function is unknown. RAGE is integrated in the cell membrane with a transmembrane spanning helix, and intracellularly consists of a C-terminal cytosolic domain for signal transduction [84,87,88]. RAGE is proposed to be a pattern recognition receptor and has many different ligands next to AGEs, among which are proteins with high inflammatory potentials such as high mobility group box 1 (HMGB1), lipopolysaccharides the signalling molecule lysophosphatic acid, damage-associated molecular patterns, and advanced oxidation protein products [83,84,87,89,90]. Which specific AGEs bind to RAGE is under much dispute. Free AGEs such as CEL and CML do not seem to bind to RAGE, whereas AGE-modified proteins and AGEs containing a peptide backbone do [86,87,91,92,93]. This suggests a possible size-dependent interaction between RAGE and its ligands, which might be due to oligomerization of RAGE in the cell membrane. Therefore, protein-bound AGEs are able to bind more strongly to oligomerized RAGE compared to individual RAGE [87]. Xue et al. (2014) found indications that in the case of MG-H1-modified peptides, the imidazolone ring of MG-H1 seems to be important in its binding to RAGE [92]. In an earlier study, we showed that blocking RAGE reduced the pro-inflammatory effect of protein bound dAGEs by almost half [73].

The large structural differences between the RAGE ligands are striking and make it complicated to find the exact elements necessary for ligand binding to RAGE.

RAGE signalling and outcome has been discussed in several reviews [84,94]. RAGE signalling leads to several different (patho)physiological outcomes including, but not limited to, inflammation, oxidative stress, apoptosis, proliferation, and autophagy [84]. Upon extracellular ligand binding activation, a small intracellular protein, mDia1, binds to the cytosolic C-domain of RAGE, activating nuclear factor kappa-light-chain-enhancer of activated B cells (NF-κB) through phosphoinositide 3-kinase (PI3K), eventually leading to the release of an array of pro-inflammatory cytokines such as Il1β, Il-6, TNF-α, and CCL2. This was confirmed by blockage of RAGE which resulted in delaying the inflammatory response [84,95]. In macrophages, ligand binding to RAGE induces a pro-inflammatory phenotype, resembling the pattern of M1 polarized macrophages, and stimulated the migration of monocytes and leukocytes through the endothelium [96]. Furthermore, RAGE signalling results in a positive-feedback loop through activation of NF-κB, which in turn upregulates RAGE expression [7,83].

In addition to RAGE, AGEs bind to a variety of different other receptors, which are mainly implicated in regulating endocytosis and degradation of AGEs. These receptors include macrophage scavenger receptors, 80 K-H phosphoprotein (or AGE-R2), galectin-3 (AGE-R3), lectin-like oxidized low density lipoprotein receptor-1 (Lox-1), fasciclin EGF-like, laminin-type EGF-like, link domain-containing scavenger receptor-1/2 (FEEL1/2), and CD36 [84]. Furthermore, AGEs bind to AGER1, also known as oligosaccharyl transferase-48 (OST48). This receptor is involved in cellular uptake and degradation of AGEs and has been found to inhibit RAGE activated NF-κB activation and to mitigate the generation of reactive oxygen species (ROS) by RAGE, as well as the downstream effects [97,98,99]. In the case of inflammatory diseases, the AGER1 to RAGE ratio seems to be decreased [100].

### 6.2. Redox Modulation and Loss of Protein Function

Paradoxically to the anti-oxidative properties, AGEs themselves also induce oxidative stress through RAGE activation. After ligand binding, RAGE activates nicotinamide adenine dinucleotide phosphate (NADPH)-oxidases, eventually leading to the formation of ROS [98]. Activation of RAGE also eventually leads to an increased intracellular nitric oxide synthase (iNOS) expression [101,102]. Next to RAGE induced ROS, AGEs themselves are formed under oxidative conditions making them very pro-oxidant molecules [103,104,105]. Due to their pro-oxidative nature, AGEs react with proteins, resulting in cross-linking or other irreversible modifications, leading to the loss of protein function [10]. It has been suggested that AGEs increase stiffness in muscle tissue, probably via cross-linking to collagen and elastin, leading to alterations in muscle function [72]. Other consequences are glycation of the blood-brain barrier, modification of LDL, impaired wound healing, and non-enzymatic glycation of DNA [106,107].

Reactions between AGEs and proteins do not only occur endogenously, but also in food products, resulting in a change of their nutritional value. Some studies have shown the loss of bioavailable amino acids in heat treated food [33]. Joubran et al. (2017) found that the MR changes the functionality of the whey protein α-lactalbumin. They also found that the specific carbohydrate involved in the reaction determined the change of the protein [29]. Another aspect of modified proteins is that they might be considered by the immune system as foreign proteins, leading to an adaptive immune response as was discussed before in the potential of dAGEs to induce allergic reactions. Kellow and Coughlan (2015) have suggested in their review that dAGEs might thereby lead to complement activation [8]. Although this hypothesis is important for the potential effects of dAGEs in the human GI system, too little evidence is known to endorse this.

## 7. Dietary AGEs and Inflammatory Bowel Diseases

When discussing inflammation in the GI tract, an important disease group to look at is IBD, consisting of, i.e., Crohn’s disease (CD) and ulcerative colitis (UC). As the name indicates, these diseases present themselves through inflammation in the GI tract. In CD patients, this inflammation may affect any segment of the GI tract from the mouth to the anus, while inflammatory patches in UC patients are limited to the colon [108]. Treatment options of IBD include the use of glucocorticoids, which are very effective anti-inflammatory drugs. However, some people are resistant to glucocorticoids. This specific glucocorticoid resistance occurs in approximately 20% of people with IBD [109,110]. IBD patients often report the influence of nutrition on their symptoms, either worsening of relieving their symptoms, causing ‘flare-ups’, which indicates a possible role for nutrition in either the onset or treatment of the diseases [4].

### Evidence of AGE and RAGE-Involvement in IBD

Several studies exist that investigated the association between dAGE consumption and IBD. Elevated levels of the dAGE pentosidine were found in several cell types within the bowel of IBD patients [111]. It was suggested that the level of pentosidine reflects the extent of tissue damage since it correlated with the expression of 8-OHdG, a marker for oxidative DNA damage. However, the pentosidine levels were correlated with age in CD, but not UC. This indicates an association between pentosidine levels and tissue damage in UC and not in CD due to this confounding factor. Additionally, the pentosidine levels were measured using an ELISA technique, which is not the most reliable technique for measuring AGEs. Andrassy et al. (2006) showed the presence of carboxymethylated (glycated) endogenous S100 proteins in the inflamed tissue of IBD patients compared to the non-inflamed tissue of the same patients [112]. Additionally, other clinical studies show evidence of a role of RAGE in IBD inflammation: In CD, protein levels of RAGE are upregulated in inflamed areas and thereby appear to play a role in the mechanisms involved in chronic inflammation. Similarly, in inflamed areas of the ileum and colon in UC, RAGE may play a role in sustaining and worsening the inflammation [113,114,115]. Furthermore, the gut endothelium in IBD patients shows a strong increase in both RAGE and NF-κB expression [20]. sRAGE, the soluble version of RAGE, is able to bind RAGE and thereby blocking the receptor. In CD patients, but not in UC patients, serum levels of sRAGE were found to be inversely correlated with CRP levels and clinical activity index [116]. Moreover, different polymorphisms of RAGE were found to be associated with IBD occurrence in different patient groups, of which at least one polymorphism significantly increased RAGE transcriptional activation [117,118]. These results indicate a potential involvement of AGEs, via RAGE, in the inflammatory processes underlying IBD.

In addition to the aforementioned human studies, several animal studies also found interplays between IBD, dAGEs, and RAGE. By exposing C57BL/6 wild-type (WT) and RAGE -/- mice to CML-modified-proteins from human inflamed intestinal tissue in the colon, Andrassy et al. (2006) showed a higher IL-6 expression in WT mice compared to RAGE -/- mice, indicating an essential role of RAGE in the intestinal inflammation [112]. Rodent studies into IBD usually involve indomethacin-induced enteritis and dextran sulfate sodium (DSS)- and 2,4,6-trinitrobenzene sulfonic acid (TNBS)-induced colitis to mimic the IBD disease status. A recent study by Body-Malapel et al. (2019) showed that RAGE -/- mice were protected from indomethacin-induced enteritis and DSS- and TNBS-induced colitis as iNOS expression was decreased in RAGE-/- mice compared to WT. This study concluded that reducing dAGEs may be a promising therapeutic target for IBD [115].

Some animal studies have been conducted into IBD and interplay with dAGEs, irrespective of investigating the role of RAGE. When giving healthy rats a heated diet that consisted of nine-fold higher dAGE levels than the normal diet, an increase in colonic oxidative stress and inflammation was seen [70]. Comparing a ‘bread crust diet’ to a normal diet, Yuan et al. (2018) found an accumulation of dietary CML in intestinal tissue in rats, with the highest accumulation in the ileum [119]. The authors attributed this accumulation to the trapping of CML in the intestinal cell, as was seen earlier in the absorption studies by Hellwig et al. (2011) and Grunwald et al. (2006) [34,35]. Additionally, a significant decrease in the activities of several antioxidant enzymes: Superoxide dismutase (SOD) and glutathione peroxidase (GSH-Px), and a significant increase in TNF-α and IL-8 secretion were found in the duodenum compared to the control group [119]. In a different study by Qu et al. (2017), rats were exposed to a regular (low-AGE) laboratory standard diet or heated (high-AGE) AIN-93G diet for 6, 12, and 18 weeks. Histological examination of the colon showed distorted crypts, crypt loss, goblet cell depletion, increased dysplasia, loose cellular arrangements, and thickening of the mucosa with oedema in rats exposed to the high-AGE diet. However, no increased inflammatory infiltration was present. The 18-week exposure group showed significantly decreased expression levels of the tight junction proteins Occludin and zonula occludens-1. All these results indicate an increased colon permeability due to the increased dAGE intake [120].

NF-κB is known to have a high activity in the GI tract of IBD patients [121]. Interestingly, transgenic mice expressing firefly luciferase under the control of an NF-κB responsive promoter showed NF-κB activation in particularly the gut after ingestion of dAGEs [122]. Remarkably, in an induced UC mice model, highly heated pellets reduced inflammation in the gut [123]. It has to be noted that this study focused on melanoidins rather than dAGEs. In this study, Anton et al. (2012) exposed mice to standard, mildly heated, or highly heated pellets, after which a reduction in histopathological damage scores in mice fed highly heated pellets compared to mildly heated and standard pellets were found. However, the differences in CML content of the mildly heated and highly heated pellets were marginal, whereas the melanoidin content was almost three times higher for the highly heated pellets. Therefore, the protective effect of the diets was probably due to the melanoidin content and not the dAGE content [123]. A follow-up study showed less mast cell infiltration in DSS treated mice when administered a highly heated diet. However, in this study the diet was not analysed on the MRP content and the treatment of the chow was different than in the aforementioned study. It is likely that the effects were again caused by the melanoidin content, but it is not possible to attribute the effects to a specific subset of the MR [124]. Additionally, ALJahdali et al. (2017) administered CML to mice for three weeks and then induced colitis in two different subgroups with either DSS or TNBS [125]. They found no inflammation caused by only CML, which can be explained by the fact that CML was administered in a non-protein bound form. They also stated that CML failed to prevent an inflammatory response, but the article did not include clear reasoning on why an anti-inflammatory effect was expected. It is unclear whether CML administration continued during colitis induction. CML did limit DSS-induced weight loss, but not TNBS-induced weight loss. They also saw a pattern towards the control state in the gut microbiota in CML-DSS exposed mice compared to only DSS exposed mice.

An important aspect in IBD is the composition and functionality of the gut microbiota. Studies into the effects of dAGEs on the microbiota in relation to IBD are scarce. An in vitro study exposed faecal samples of healthy volunteers vs. UC patients to native bovine serum albumin (BSA) and glycated-BSA (with glucose) [126]. When exposing the microbiota from healthy volunteers to glycated-BSA, the colonic microbiota reflected that of UC patients. In both UC and non-UC microbiota, an increase in sulphate reducing bacteria was seen when exposed to glycated BSA versus native BSA. Within the UC samples, a significant increase in the number of *Clostridium perfringens/histolyticum* group and *Bacteroides* spp. and a significant decrease in the number of bifidobacteria and *E. rectale* group was seen after exposure to glycated BSA compared to native BSA. These changes were not seen in the control model (non-UC subjects). No changes in the SCFA concentrations that could be attributed to any treatment were seen in the models [126]. This indicates that dAGEs may contribute to some of the symptoms of IBD.

From the results of the studies investigating the interplay between IBD, RAGE, and dAGEs it can be concluded that RAGE is involved in the symptoms and possibly in the disease course of IBD. This makes decreasing the levels of the food-derived natural ligands of RAGE (i.e., dAGEs) an interesting therapeutic option. However, more research is needed in humans to validate this relationship.

## 8. What Are the Characteristics of AGE-Rich Food Products?

### 8.1. Factors Influencing the MR

The MR and other pathways through which dAGEs form, can be influenced by different factors. An important factor influencing the rate of dAGE formation is the macronutrient composition of the food products. The presence of sugar leads to both initiation of the MR and caramelization. Proteins are made up from amino acids, so their presence increases the rate of the MR as well. Lipids can be oxidized, leading to the formation of 1,2-dicarbonyls via the Namiki pathway [10]. One factor relating to this is the reactivity of the amino groups themselves. Primary amines are more reactive than secondary amines, whereas tertiary amines are inactive [27]. The primary amino group present in the side-chains of lysine is the most reactive precursor amine in proteins. Next to this, the guanidino group of arginine, the indole group of tryptophan, and the imidazole group of histidine are reactive sites, but to a lesser extent. Practically any N-terminal amino group and free amino group is reactive [127].

The sugars that react in the MR are known as reducing sugars. These are sugars with a free aldehyde or ketone group such as glucose, galactose, and fructose [128]. In addition to these groups, other chemical characteristics of sugars, carbohydrates, and amino acids also determine their reactivity in the MR. In a very early study by Ashoor et al. (1984), the browning of common amino acids in combination with different sugars was assessed and the different amino acids were categorized by high, middle, and low reactivity [129]. The high reactivity group included Lys, Gly, Trp, and Tyr. The middle group included Pro, Leu, Ile, Ala, Hypro, Phe, Met, Val, Gln, Ser, and Asn. The amino acids with the lowest reactivity in their study were His, Thr, Asp, Arg, Glu, and Cys [129]. Although the high reactivity of lysine is shown in several studies, the level of reactivity of the other amino acids is under dispute [127,130,131,132]. Kwak et al. (2004) analysed the browning of all possible combinations between 12 different amino acids with five different sugars (glucose, fructose, maltose, xylose, arabinose), which is an indication of MRP formation. Lysine showed the highest formation of MRPs with all five sugars. The order of reactivity of the other amino acids depended on the type of sugar: Pentoses were for instance more reactive than hexoses [131]. Most commonly analysed and used dAGEs are therefore based on lysine and arginine. In the case of glucose, arginine was indeed the second most reactive amino acid. However, the reaction between glucose and arginine did not lead to a brown colour, despite the fact that glucose was depleted quickly.

By using a peptide library (SPOT library) to assess the reactivity of different amino acid residues to glucose, fructose, and AGE-BSA, Münch et al. (1999) also found that the binding reactivity of amino acids depended on the type of sugar or dAGE provided [127]. In the case of glucose and fructose, amino acids with nucleophilic side chains (lysine, cysteine, and histidine) were prone to bind sugars, whereas AGE-BSA preferentially bound to arginine and tryptophan and to a lesser extent to lysine. This difference in binding reactivity between dAGEs and reducing sugars could be an explanation why both lysine and arginine based protein-bound dAGEs are largely present in food products.

To identify the products formed during the MR, Hemmler et al. (2018) recently used a non-targeted analysis with Fourier transform ion cyclotron resonance mass spectrometry (FT-ICR-MS) to derive the molecular formulae of compounds formed after 10 h of heating at 100 °C for different sugar-amino acid combinations [132]. The reaction rate order that was found for amino acids was: Lysine > cysteine > isoleucine ≈ glycine. Interestingly, when assessing the browning of the different combinations, the order of reactivity was lysine > isoleucine > glycine > cysteine, showing that cysteine produces many non-brown MRPs. The order of reaction rates for the sugars was: Ribose > arabinose > fructose ≈ xylose > galactose > glucose. Joubran et al. (2017) confirmed the higher galactose reactivity compared to glucose in combination with α-lactalbumine, which can be explained by the equatorial orientation of the hydroxyl groups in glucose and glucose’ stable closed-ring structure. The open chain-form of sugars react more quickly than the closed ring [29]. Important in the sugar reactivity is also the proportion of free carbonyls and the length of the carbon backbone. As seen earlier in the study of Kwak et al. (2004), pentoses have a faster reaction rate than hexoses. Next to this, the monosaccharides have a higher reactivity than disaccharides, with disaccharide being more reactive than oligosaccharides [29,133].

Next to the composition of the raw materials, baking conditions influence the formation of dAGEs to a large extent [134]. An increase in temperature leads to an increased formation of dAGEs. This increase can be reached through baking, cooking, roasting, and microwave heating. The compounds formed during the different pathways are very reactive, but for the reaction to continue the molecules must react with each other. Therefore, diluting these molecules in aqueous solutions slows down the formation of dAGEs. Moreover, H^+^ ions are capable of binding to the intermediate reactants, making them less reactive and slowing down the formation of dAGEs. These effects have nicely been reviewed by Lund et al. (2017) [133].

The MR is an ongoing reaction and therefore does not only occur during the processing of food products, but also during the storage of food. Storage conditions apparently influence the rate of the MR, such as temperature and moisture/humidity level [135]. Since 1,2-dicarbonyls can be formed via the Namiki pathway from the oxidation of fat, the presence of oxygen in the packaging can accelerate the MR, while the addition of antioxidants may hamper the rate of the MR due to trapping of oxidation products. A practical example of the importance of starting product in the MR is the difference between goat’s milk infant formula and cow’s milk infant formula. Infant formulas from goat’s milk have a lower CML content than from cow’s milk, consequently also heat processing leads to lower levels of CML in goat’s milk compared to cow’s milk [136].

### 8.2. Assessment of Dietary AGEs in Food Products

In order to define the characteristics of dAGE-rich food products, it is important to know in what kind of food products they are present. This brings us to an important problem in the current literature. Since dAGEs are a heterogeneous group of compounds, it makes it difficult to analyse them very specifically. Different methods are being used throughout the literature to determine the amount of dAGEs present in different food products. Several AGEs have fluorescent properties and the fluorescence spectrum of vesperlysine (λ_ex_ = 370 nm/λ_em_ = 440 nm) is considered as an AGE-wide specific fluorescence in multiple studies [137,138,139]. A major drawback of this method is that not all AGEs have fluorescent properties. For instance CML, which is regarded in the current literature as the most important AGE in food, does not have any fluorescent properties [140]. A different technique used in multiple studies to detect dAGE presence is competitive ELISA [141]. Both ELISA and fluorescence assays are easy to perform but have a low accuracy in measuring dAGEs. As stated before, dAGEs are a group of different compounds and for ELISA you need specific antibodies for specific molecules. Goldberg et al. (2004) made a database of food products measured with an AGE ELISA using anti-CML antibodies [141]. However, it is questionable what specific molecules are recognized by the anti-AGE antibodies. The values obtained in these studies are expressed as AGE-kU, for which the definition of AGE-kU remains unclear. Therefore, it is only possible to compare trends between food products within the same study.

More specific analytical techniques are based on chromatography and mass spectrometry. UPLC-MS/MS can more specifically measure specific molecules and quantify them. Other, less often used, techniques are nuclear magnetic resonance (NMR) spectroscopy [142] and size exclusion chromatography combined with fluorescence [143]. All methods used in the analysis in MRPs have been recently reviewed by Troise [144].

Interestingly, the use of different techniques leads to different results. Figure 2 depicts the average CML content of different food groups per average portion size, comparing the results of the ELISA database of Goldberg et al. (2004) and the mass spectrometry database of Hull (2012) [141,145]. As can be seen, the trend between the two methods regarding the different food groups is completely different. This was also seen by Niquet-Léridon et al. (2015), who showed lower CML levels and other CML patterns when comparing food groups, especially in the food groups high in fat and carbohydrates, when measured by ELISA compared to the LC-MS/MS measurement [146].

Several databases exist on the amount of dAGEs in food products measured by mass spectrometry. Scheijen et al. (2016) assessed CML, CEL, and MG-H1 levels in the Dutch food products [147], Delatour et al. (2009) assessed CML in dairy products [148], and Hull et al. (2012) assessed a vast range of food products from the Western diet on CML levels [145]. The precursors 3-DG, 3-deoxygalactosone, MG, and HMF have been assessed by Degen et al. (2012) [149]. Table 2 shows a selection of food products relatively high in dAGE content, based on the database published by Scheijen et al. (2016).

Additionally, although the databases published give good indications on the levels of specific dAGEs in food products, they do not make a distinction between free and protein-bound dAGEs. Protein-bound dAGEs are in general much more abundant in food products than free dAGES, as was recently reviewed by Zhao et al. (2019) [150]. The ratios in which dAGEs are present in either free or protein-bound form seems to depend on the type of food product, as was shown in the study by Hegele et al. (2008) who compared levels of free and protein-bound dAGEs in raw and processed milk products [151]. In samples of pasteurized milk, especially the early glycation product N^ε^-fructoselysine was found to be up to 800-fold higher present in protein-bound form compared to free form, protein bound CML was found in levels 30-fold higher than its free form. Levels of dAGEs in pasteurized and ultra-high temperature (UHT) treated milk did not differ significantly from levels found in raw milk. However, increasing the processing led to much higher levels of dAGEs as in the case of condensed milk. Processed food proteins generally have 2% of their lysine modified. However, in samples such as heated milk and bread crust this can be up to 10–20% [152].

### 8.3. Exposure to dAGEs

The results of the available literature show that dAGEs formed during the MR have diverse effects on human GI health, depending on the way of food processing and the type of food product. The question remains to what extent humans are exposed to dAGEs. Most of these databases present on the dAGE content in food products measure only the presence of CML in food products. Several complications occur when assessing the intake of dAGEs using a database. Firstly, many food products have to be prepared, and the method of food preparation largely determines the levels of dAGEs. Some people eat their steaks ‘raw’ while others cook it for a longer time and enjoy theirs ‘well done’. However, the effects of these preparation ways are not included in the databases. Secondly, Niquet-Leridon et al. (2015) showed large differences in CML content between different brands of the same product. This might be due to the use of different starting products, thermal treatment, and packaging [146]. Thirdly, as discussed before, many different dAGEs exist, and assessing the amount of one dAGE in a food product most likely does not give a proper representation of the total amount of dAGEs in the food product. Therefore, Zhu et al. (2018) proposed a panel of lysine-, arginine-, cysteine-, and nucleotide-derived dAGEs to estimate the amount of dAGEs in food products in a more accurate way [153]. Lastly, many of these databases make no distinction between protein-bound and free dAGEs, while our results show that this distinction is important to assess the possible effects. For future studies into the exposure of humans to dAGEs, it is therefore pivotal that a consensus should be made on the measuring techniques of dAGEs and to acknowledge the problem that occurs with different food processing techniques.

Even though some concerns exist on the currently available dAGE databases, it is still interesting to have an indication on the contribution of specific food products to the daily dietary intake of dAGEs. Scheijen et al. (2016) analysed many commonly consumed Dutch food products on the presence of CML, CEL, and MG-H1 [147]. We calculated the intake of three dAGEs combined (CML, CEL, and MG-H1) per day per food item of the particular food item users from 19–30 years old in the Netherlands, based on the database published by Scheijen et al. (2016) and the Dutch National Food Consumption Survey [154]. Table 3 shows some examples of the food products within the top 15% range of the amount of dAGEs/day. The amounts of CML, CEL, and MG-H1 were summed up to give an overview on the total amount in the product.

### 8.4. Can This Be Clarified by the Maillard Reaction?

When looking at the top 15% of food products that contribute to the daily amount of ingested dAGEs in Table 3, all products are subjected to thermal treatment during processing. Blood sausages, steak, and meat balls are all meat products containing proteins and sugars, as well as lipids that can contribute to the MR through the Namiki pathway. The cereals depicted are toasted corn topped with sugar syrup, leading to a faster reaction rate than normal cereals. Peanuts also contain 52% fat, 26% protein, and 3.1% sugars and are roasted at high temperatures. Brown bread has a relatively low amount of dAGEs per 100 g compared to the other products, but the high consumption leads to a relatively high contribution to the total amount of dAGEs ingested per day. In general, the relatively high levels of dAGEs in these food products can therefore be clarified by the MR. It is also important to note that reheating of food products leads to additional dAGE formation, since the rate of MR will be accelerated by the reintroduction of heat to the food product.

### 8.5. Populations at Risk

Using the different dAGE databases and food consumption data from different population groups, it might be possible to distinguish groups of people that eat a lot of dAGEs and thus to define which people would be at higher risk for the intestinal effects of dAGEs. However, the effect of dAGEs on the intestinal health of healthy people is currently unknown, and more research is needed to be able to use this approach for the general population. In contrast, in this review we have shown that people who already have an inflamed GI tract, such as IBD patients, may experience negative effects from dAGEs. Although scientific evidence is scarce, diet is known to play a role in the symptoms of IBD. Commonly given diet-related advice for IBD patients includes avoidance of processed meat, fried foods, sugars, and nuts [4,155]. Interestingly, exactly these products are also listed in our exposure assessment in Table 1 and thus contain a lot of dAGEs. Based on the pro-inflammatory effect of dAGEs seen in this review, it seems logical that patients suffering from IBD will need to avoid these products as they may benefit from eating less dAGEs.

In addition to a possible correlation between the exposure to certain food groups and the presence of IBD symptoms, another important aspect in IBD is the composition and functionality of the gut microbiota. The studies assessed in this review indicate that dAGEs may influence the already impaired microbiota of IBD patients.

Irritable bowel syndrome (IBS) is a gastrointestinal disorder with frequent abdominal pain and changes in stool frequency and form, but no known underlying pathology or aetiology [156]. However, increased intestinal permeability and presence of activated immune cells in the GI tract have been found in IBS patients [156,157]. Similar to IBD, IBS patients often report the influence of different food products on their symptoms. Moreover here, these symptoms could be influenced by the extent to which patients are exposed to dAGEs. Based on the results from the discussed studies, any person suffering from a disease or a drug side-effect that influences the GI epithelial layer or leads to macrophage presence in the GI tract, may suffer from consequences of a high dAGE diet. In addition to IBD and IBS, these disorders include, but are not limited to: Intestinal ischemia, chemotherapy-induced mucositis, NSAID-induced enteropathy, intestinal infections, celiac disease, food allergy, and heartburn [158]. As discussed before, low gastric pH is important to protein digestion and the dAGE-protein binding is important for the pro-inflammatory effects. Antacids neutralize the gastric acid to prevent heartburn, which might lead to even more protein-bound dAGEs in the GI tract [159].

Additionally, the use of specific food products as part of a specific diet may lead to a higher dAGE intake. For instance, lactose-free UHT milk may contain more dAGEs due to lactose being hydrolysed into glucose (reacts 10 times more rapidly than lactose) and galactose (reacts 20 times more rapidly than lactose) [133]. Finally, infants are a sensible group when it comes to GI health. During infancy, the GI tract and the immune system are still developing and have an important function in the development of allergies [79,80]. Compounds influencing the GI tract such as dAGEs may have detrimental effects on the GI tract of infants. Infant formulas contain protein-bound dAGEs due to the high processing temperatures [160]. A study from 2008 showed 28 to 389 times more CML in infant formulas than breastmilk [44]. Hydrolysed infant formulas also contained significantly more CML than the non-hydrolysed formula. Additionally, breast–fed infants showed approximately 46% lower plasma CML levels compared to the formula–fed infants, showing the uptake of CML into the systemic circulation in infants, although the majority of this CML was excreted via urine [44].

## 9. Conclusions

In this review, we investigated to what extent dAGEs may cause a detrimental effect in the human GI tract. From the available studies, we can conclude that dAGEs need to be protein-bound to exert a pro-inflammatory effect and that decreased digestibility of glycated proteins lead to a presence of protein-bound dAGEs throughout the complete GI tract. Additionally, dAGEs are formed in the GI tract. Once present in the GI tract, they are likely to exert different effects, especially in IBD where RAGE seems to be upregulated. Although more clinical trials are needed and to our knowledge, a direct association between dAGE-intake and IBD symptoms has not been investigated yet. We additionally have pinpointed several pitfalls in dAGE research. Most notably, due to the formation of many different compounds during the MR, many effects seem to be food type and thermal treatment dependent. This is an important aspect that is currently not being taken into account by many dAGE-researchers.

## Figures and Tables

**Figure 1 nutrients-12-02814-f001:**
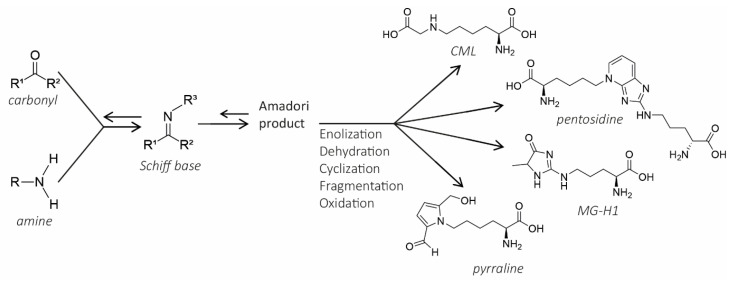
Simplified overview on advanced glycation endproducts (AGEs) formation via the Maillard reaction (MR). As an example, the molecular structure of four different dietary (dAGEs) are shown. CML: N”-carboxymethyl-lysine; MG-H1: methylglyoxal-derived hydroimidazolone-1.

**Figure 2 nutrients-12-02814-f002:**
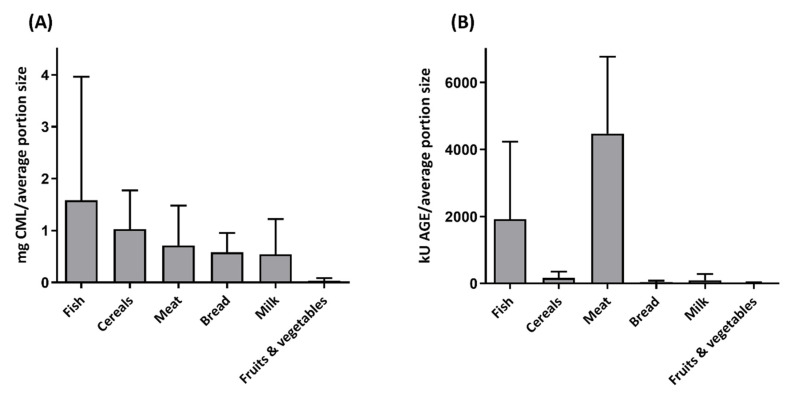
In the AGE content of food products using different quantification methods. (**A**) Data from Hull et al. based on mass spectrometry analyses [145]. (**B**) Data from analyses using N^ε^-carboxymethyl-lysine (CML) antibodies from Goldberg et al. based on ELISA analyses [141].

**Table 1 nutrients-12-02814-t001:** Summary of uptake of individual dAGEs in the in vitro studies.

Compound	Evidence	Notes	Reference(s)
Nε-carboxymethyl-lysine (CML)	Strongly retained inside Caco-2 cells.	Diffused into gastrointestinal epithelial (Caco-2) cells, but was not able to cross the basolateral membrane. Accumulation in intestinal cells. Not likely transported by amino acid and peptide carriers and the transepithelial flux measured for the compounds occurs most probably by simple diffusion.	[34,35]
Nε-carboxyethyl-lysine (CEL)	Strongly retained inside Caco-2 cells.	-	[34]
methylglyoxal-derived hydroimidazolone-1 (MG-H1)	Strongly retained inside Caco-2 cells.	-	[34]
Maltosine	Absorbed as dipeptide into Caco-2 cells by PEPT1 and strongly retained in cells. Not absorbed in free form.	Free maltosine permeates the basolateral cell membrane by simple diffusion down its concentration gradient and possibly by the action of basolateral amino acid transporters.	[34,36]
Glycated dipeptides	Absorbed into Caco-2 cells by PEPT1.	Intracellularly hydrolysed by peptidases to the free modified amino acids and alanine.	[34]
Pyrraline	Not free pyrraline, but the dipeptide with alanine is absorbed by PEPT1 in HeLa cells (cervical cancer cells).	After intracellular hydrolysation free pyrraline diffused through the basolateral membrane.	[34,37]
fructoselysine	Simple diffusion to a small extent in Caco-2 cells.	Not likely transported by amino acid and peptide carriers and the transepithelial flux measured for the compounds occurs most probably by simple diffusion.	[35]

**Table 2 nutrients-12-02814-t002:** Average N^ε^-carboxymethyl-lysine (CML), N^ε^-carboxyethyl-lysine (CEL), and methylglyoxal-derived hydroimidazolone-1 (MG-H1) content of several food products as published by Scheijen et al. (2016) [147].

Food Product	CML (mg)/100 g	CEL (mg)/100 g	MG-H1 (mg)/100 g
Blood sausages	4.8	7.7	63.0
Peanut butter	3.1	6.7	44.5
Cereals	2.0	1.6	41.6
Ginger biscuit	2.5	2.0	28.3
Salted peanuts	1.7	3.4	25.7
Rusk	2.0	1.4	23.1
Red cooked beef	2.0	5.6	13.5
Chocolate sprinkles	5.1	2.0	9.3
Canned salmon	1.2	2.8	11.0
Fried tofu	0.9	1.2	10.9

**Table 3 nutrients-12-02814-t003:** Average intake of three dAGEs combined (CML, CEL, and MG-H1) per day per food item of the particular food item users from 19–30 years old in the Netherlands.

Food Product	dAGEs Content (mg/100 g) [147]	Average Intake per Day (g)	Daily Exposure to dAGEs (mg/day)
Blood sausages	75.5	32	24.2
Beef steak (canned)	18.7	78	14.6
Cereals (frosted flakes)	27.1	39	10.5
Fried rice	10.9	91	9.9
Peanut butter (Calve)	51.5	18	9.3
Brown bread	6.7	100	6.7
Peanuts	31.7	21	6.7
Meat ball	10.1	60	6.1
Chicken Wings	4.6	127	5.9
Ginger biscuit	32.8	12	3.9

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
