# Peer review of "Dietary Advanced Glycation Endproducts and the Gastrointestinal Tract"

_nutrients, 2020, doi:10.3390/nu12092814_

Round 1

Reviewer 1 Report

The review work by Timme van der Lugt, Antoon Opperhuizen, Aalt Bast and Misha F. Vrolijk entitled Dietary advanced glycation endproducts and the gastrointestinal tract is informative. This review work has covered many aspects of AGEs. If the below points are further discussed, it will make this work more complete. 

  • 4. Dietary AGEs in the gastro-intestinal tract: digestion, absorption, formation and degradation
    • Are there any studies reporting about how many % of AGEs can be absorbed and how long AGEs can remain in the GI tact (retention time in GI tract)?
  • 8.1. Factors influencing the MR
    • L599-L637: The relationship between MR and types of amino acids and sugars has been discussed. How those information could be useful for general readers? It would be more interesting and useful to interpret this information, such as what kinds of foods that consumers should avoid?
  • Figure 2. (A)Mass spectrometry analysis and (B) ELISA analysis should be indicated in the figure directly or in the legend. 
  • At the present, are there any studies or references that can be used to draw a suggestion of Upper Intake Levels (per day) of AGEs that still can be safe or not cause harmful consequences for consumers?
  • Can using microwave to warm up or cook food contribute to more AGES? And can using microwave be a reason for increasing the incidence of IBD in industrialised or developed countries? Actually, IBD is rare in Asian countries and and regions, but recently the incidence has been increasing in Japan, Taiwan, and China.  

Reviewer 2 Report

The authors present a comprehensive review assessing the effects of dietary advanced glycation endproducts (dAGEs) on the GI tract and GI health. The review includes an extensively detailed presentation of the biochemical aspects of dAGEs, but relatively less informative physiological impact of dAGEs/other MRPs on the pathogenesis of IBD and how it may influence future therapeutic ventures (speaking from a clinician's perspective). This is likely to be due to the palpably scant published literature linking dAGEs to IBD pathogenesis. However, given the amount of detail in the overall structure of the review, I commend the authors for their hard work. The authors have paid careful attention to support the science presented through adequate referencing. Therefore, I do not have any specific questions or suggest corrections. 

Reviewer 3 Report

This Review is very interesting and appropriate

The authors have done an excellent job about a difficult subject and have put some clear and brilliant ideas to know the actual state of art. The strong and weak points about the research field and the need to continue doing more studies, especially in humans

The role of the AGE in the health intestine and in situations of chronic inflammation such as occur in IBD patients is analyzed and the future use as prevention and also possible treatment of these diseases is remarked

I have included in my review some suggestions to the authors in order to clarify some points

Review Comments : Nutrients 929429

Q1 : Title

Is correct

Q2 : Abstract

The first phrase, talking about “the prevalence of IBD in the world” can be erased.

Into the keywords the term “inflammatory bowel disease” also can be erased, because the focus of this paper is different

Q3 : Introduction

Lines 26-36, can be eliminated.

It would be better start in line 37, taking about the “Western diet typically includes….”

The Figure 1 must be placed down the introduction section on the paragraph talking about the Maillard reaction at the end (Line 102)

Q4 : Terminology

On line 109, the acronymus of CML, CEL, MG-H1 must be explained before presented

4.1. Digestion of AGEs

This Section is too long. Must be shortened if possible.

4.2 . Absorption

In Table 1, the meaning of CML, CEL, and MG-H1, must be specified and named

4.3. Formation of AGEs in the GI tract

On line 242, please explain the meaning of TIM-1

4.4. Degradation and effect in the colon

The conclusions are clearly exposed.

On line 328, you can eliminate the final word of conclusions in order to avoid redundance

Q5. Local health effect in the intestinal tract (please include the, before intestinal)

5.1. Pro-inflammatory effects of dAGEs.

Is OK

5.2. Other local effects

The final paragraph is well explained

  1. Mechanisms

6.1. Receptor mediated effects.

Line 419. Please change the adjective “most” by “better”, because is more appropriate and explain the meaning of RAGE for first time

Line 425, please erase superfamily because is repeated

6.2. Redox modulation and loss of protein function

Line 469. Please explain the meaning of ROS because is the first citation of this abbreviation

Line 470. Please do the same with iNOS

  1. Dietary AGEs and Inflammatory Bowel Diseases.

7.1. Evidence of AGE and RAGE involvements in IBD

It would be convenient to add this phrase at the end of this section :

“There is not a clear relationship especially in humans and is neded to wait for more and new findings in future studies”

  1. What are the characteristics of AGE-rich food products?

8.1. Factors influencing the MR

This section in nicely described and completed

8.2. Assessment of dietary AGE in food products

As we can see in Figure 2, there are great differences in the AGE content of food products and great variability in every component.

Please can you give some explanations for these discrepancies

In Table 2 add a legend explaining the meaning od CML, CEL and MG-H1 for better comprehension for the readers

8.5. Populations at risk

There is confusing information about age and intestinal damage.

Please can you clarify better these effects along children, adults and elder people

  1. Conclusions

The main conclusion is that the MR effects seem to be food-type and thermal treatment dependent

  1. References

Are good enough, recent and great number (160)

Reviewer 4 Report

The paper is well designed and quite well written. Language polishing should be performed by native speaker.
